# Wealth-based inequalities in early childhood development (ECD) outcomes in Bangladesh: A decomposition analysis using MICS 2019

Tasfia Tasneem Ahmed[1], Nafis Sadik [2]*

1 Department of Finance, Bangladesh Institute of Governance and Management (BIGM), Dhaka, Bangladesh, 2 Research Wing, Bangladesh Institute of Governance and Management (BIGM), Dhaka, Bangladesh

* nafissadik72@gmail.com

## Abstract

Early Childhood Development (ECD) inequality among different socioeconomic groups is a rising concern in developing countries. This paper aimed to identify and decompose the ECD inequality among poor and non-poor groups in Bangladesh. For measuring inequality in ECD, concentration curves and their corresponding indices were used in this study utilizing the Multiple Indicator Cluster Survey 2019 data. Furthermore, a standard decomposition approach has been used to decompose the inequality. Findings reveal that children from poor families consistently exhibit lower scores across all ECD domains. Notably, a significant disparity exists in the literacy-numeracy domain, with a concentration index of 0.1825. A lower proportion of poor children (34.17%) meet developmental milestones compared to their well-off counterparts (65.83%). Key determinants influencing ECD outcomes among the poor group include attendance at early childhood education (ECE) programs, sex of the child, multiple childcare involvement of mother, and supervision quality. Additionally, wealth-based disparities in ECD outcomes can be attributed to factors such as maternal education, access to books, nutrition, and the quality of childcare provided. The findings underscore the urgent need for targeted interventions and policy reforms aimed at addressing the specific needs of disadvantaged children, especially for literacy and numeracy skills. By prioritizing poverty reduction initiatives, access to quality Early Childhood Education (ECE) programs, provision of books, improving maternal education, and enhancing supervision quality, efforts can be made to narrow the wealth-based gap in ECD.

## Background

Early childhood development (ECD) is a crucial predictor of long-term health and well-being in children and includes their cognitive, motor, language, and socioemotional development [1–5]. It is increasingly acknowledged as a significant influencer

**Data availability statement:** Publicly available data were used which is accessible from the UNICEF MICS website (https://mics.unicef.org/surveys). The Multiple Indicator Cluster Survey (MICS) 2019 is a publicly available dataset collected by the Bangladesh Bureau of Statistics (BBS) in collaboration with UNICEF. The MICS is an internationally recognized household survey program designed to provide statistically sound and internationally comparable estimates on key indicators related to the health, education, and well-being of women and children. No special privileges were granted to the authors in accessing this data, and all researchers can freely access the original dataset from the UNICEF MICS website.

**Funding:** The authors received no specific funding for this work.

**Competing interests:** The authors have declared that no competing interests exist.

**Abbreviations:** ECD, Early Childhood Development; ECE, Early Childhood Education; ECDI, Early Childhood Development Index; GER, Gross Enrollment Rate; MICS, Multiple Indicator Cluster Survey; OR, Odds Ratio; RC, Reference Category; SDG, Sustainable Development Goals; WHO, World Health Organization.

shaping the trajectory of an individual's future development [6]. Healthy early development provides the foundation for people's success in school, their overall health, secure communities, effective parenting, and the stability and economic development of societies [7]. It is a period where a process of synaptic pruning mostly takes place which continues into early adulthood [8]. Investments in early childhood development yield significant returns, contributing to stronger societies and economically vibrant futures. Due to the immense importance of ECD, it is acknowledged in the United Nations' Sustainable Development Goals (SDG). According to Goal 4.2 of the SDG standards, member nations must guarantee that all children have access to high-quality early childhood development by 2030 for them to be prepared for primary education [9].

Despite the immense importance of early childhood development, many socioeconomic groups, especially in developing countries are lagging in ECD [10–12]. Children from low-wealth families frequently face unfavorable outcomes for their infants, including low birth weight and preterm birth, which are linked to cognitive deficits and further impede the development of the child [13]. Lack of fundamental investments and stimulation during childhood can cause developmental deficiencies in children from low-income families that are difficult to correct without intervention [14]. Studies have shown that socioeconomic circumstances during infancy and early childhood significantly shape developmental opportunities, with long-term implications for intergenerational poverty transmission [15,16]. A significant portion of the disparities in children's early development can be attributed to differences in their upbringing, schooling, emotional health, and material surroundings [17]. Children's cognitive growth and behaviour are greatly impacted by family income and poverty status. Children living in wealthy neighborhoods tend to have higher IQs, whereas children living in low-income neighborhoods are more likely to have externalising behaviour issues [18].

As early childhood development is crucial for economic growth, understanding differences across socioeconomic groups is crucial for policymakers and child well-being [19]. Among these groups, those with limited wealth and financial resources are disproportionately affected, facing heightened vulnerability in early childhood development outcomes [16,20]. The situation is particularly alarming in South Asia with household wealth inequalities in the ECDI being highest in South Asia compared to other global regions [11]. Because of their poor living conditions, 89 million children in South Asia under the age of five are at risk of not developing to their full potential [21]. On the same note, a higher socioeconomic position is linked to improved ECD in this region [22].

In light of the above context, Bangladesh, as a country in South Asia, is a good case study to analyse the wealth-related inequality in ECD. Various studies have outlined Bangladesh to be in a disadvantageous position in ECD [8,23,24]. Bangladesh is one of the ten nations that house the most impoverished children at grave risk of delayed cognitive and social-emotional development, according to a report by World Health Organization (WHO) [25]. Bangladesh's preprimary gross enrollment rate (GER) is below average for South Asian nations and below average when compared

to higher-income and upper-middle-income nations [26]. Moreover, Bangladesh has little state spending on early childhood education (ECE) and poorer households spend far less on ECE than do wealthier households [26].

Recent literature on ECD in Bangladesh has mostly focused on factors determining early childhood development [8,24,27–31] The results of the studies have indicated that early childhood development and wealth status are positively correlated, among other things. Richer households had a higher likelihood of attaining ECD status for their children than did poorer households [8,24,27,28,31].

However, there is a scarcity of systematic evidence to assist policymakers and donors in assessing the degree to which disadvantaged children in Bangladesh are falling behind in early childhood development. In this situation, a deeper understanding of the root causes of the wealth inequality dimension of early childhood development in Bangladesh has policy relevance. It is essential to develop targeted interventions and policies that can help bridge the gap. Moreover, previous studies mostly focused on inequality in children's physical health rather than neurodevelopment. Limited research has looked into children's early childhood development with a concentration index and decomposition approach [32,33]. In this case, concentration curves and decomposition analysis provide a useful tool for dissecting the disparities in early childhood development and identifying the specific contributors to these inequalities. By uncovering the underlying factors that drive inequities in ECD, policymakers and practitioners can design more effective mechanisms to guarantee that all children, irrespective of their socioeconomic status, can unlock their true potential.

To the best of our knowledge, no study has yet systematically decomposed the ECD gap in Bangladesh into components with a clear contribution from different factors. Using data from MICS6 2019 Bangladesh, we provide a decomposition of wealth-based inequalities in ECD using available data in four domains of ECD. This data provides unique information to monitor levels of ECD attainment and inequality in Bangladesh. The evidence obtained from this study will help to design effective interventions for low-wealth families so that ECD inequality can be narrowed and all children can meet their development potential.

This paper contributes to the literature in three notable respects. First, it examines wealth-based inequality across all four domains of ECD, literacy-numeracy, physical, socio-emotional, and learning, using concentration indices and concentration curves. Second, it applies a standard decomposition method to quantify the contribution of specific socioeconomic and demographic factors to the observed inequalities. Third, by analyzing domain-specific disparities, the study offers targeted insights for designing policy interventions to improve ECD outcomes and reduce inequities in Bangladesh. This disaggregated approach is a significant advancement in the literature, as prior studies have primarily focused on overall wealth-related ECD inequality, overlooking how different dimensions of early childhood development may be unequally affected by socioeconomic status.

Understanding how different dimensions of early childhood development are unequally affected by socioeconomic status is essential for designing more effective and equitable interventions. For instance, if wealth inequality is particularly affecting literacy and numeracy skills, interventions can focus on providing low-income families improved access to early learning materials, affordable and quality preschools, along with support for parents to help children develop these fundamental skills at home.

## Materials and methods

### Data and sample

For this study, we have used the Bangladesh Multiple Indicator Cluster Survey (MICS) conducted in 2019 (S1 Data). The survey was carried out by the Bangladesh Bureau of Statistics (BBS) in partnership with UNICEF Bangladesh as part of the Global MICS Programme. This survey collected data from 61,242 households across all 64 districts [23].

UNICEF developed the Global MICS Programme in the 1990s as an international multi-purpose household survey program to assist countries in gathering internationally comparable data on a wide range of child and women's welfare indicators. MICS surveys assess key indicators that enable governments to create data for use in policies, programs, and

national development plans, as well as to track progress toward the Sustainable Development Goals (SDGs) and other internationally agreed-upon objectives. For this study, we have used data from children aged 36–59 months. The schematic diagram for sampling design is showed in Fig 1.

## Variable definition and measurement

**Outcome variable.** The outcome variable of the study is the ECD status of children aged 36–59 months. Early childhood development (ECD) encompasses a variety of dimensions and entails a sequential advancement of motor, cognitive, language, socio-emotional, and regulatory abilities during the initial years of life [6]. Crucial facets of a child's holistic growth include physical development, literacy and numeracy aptitudes, socio-emotional development, and preparedness for learning. These domains lay the groundwork for future endeavors and establish the course for health, educational attainment, and overall well-being throughout life [34].

UNICEF developed the Early Child Development Index (ECDI) as a tool to measure and monitor progress in early childhood development globally [35]. A 10-item module is utilized to compute the ECDI, which serves as a tool to assess the developmental status of children in countries. This index relies on specific developmental milestones expected to be attained by children aged 3 and 4. These 10 items are employed to evaluate if children are progressing developmentally across four domains: Literacy-numeracy, Physical, Social-emotional and Learning.

For the "Literacy-numeracy" domain, children's developmental progress is determined based on their ability to recognize a minimum number of letters of the alphabet, read at least four simple, common words, and identify

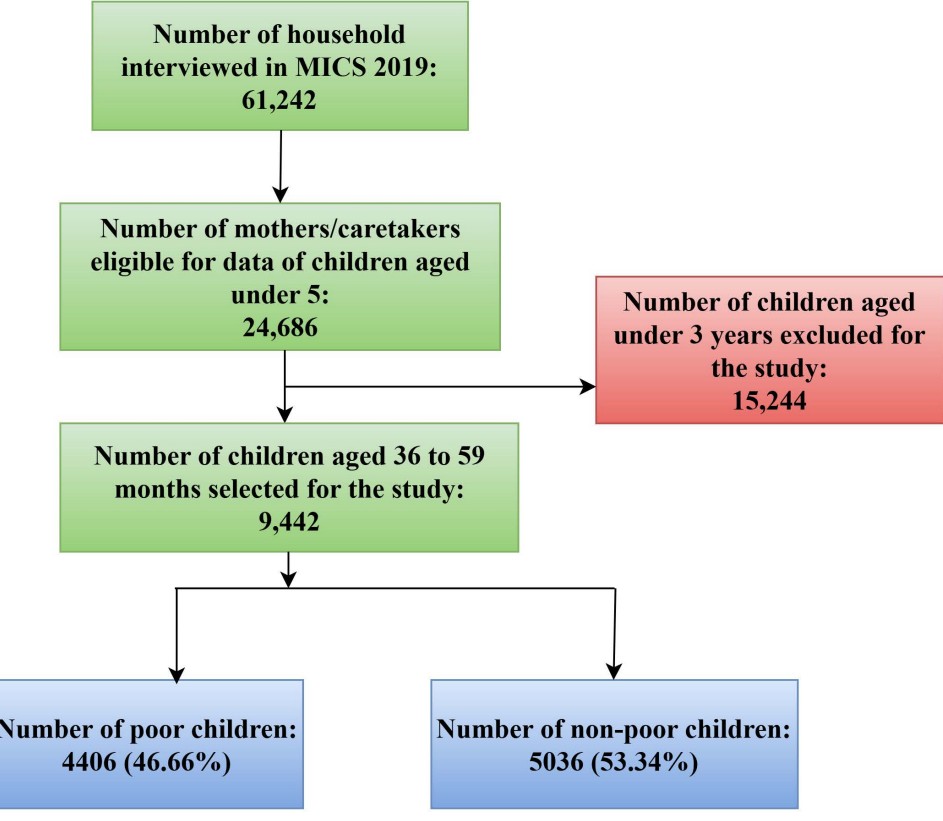

**Fig 1. Schematic diagram for sampling design.**

the names and symbols of numbers 1–10. Achievement in at least two of these areas qualifies a child as developmentally on track in this domain. For the "Physical" domain, a child is deemed developmentally on track if they can grasp a small object with two fingers, such as a stick or a rock from the ground, and if the mother/caretaker does not report the child being occasionally too unwell to engage in play. In the case of "Social-emotional" domain, children are considered to be developmentally on track if at least two of the following statements are true: the child interacts well with other children, refrains from aggressive behaviors such as kicking, biting, or hitting, and demonstrates minimal susceptibility to distraction. Lastly, for the domain of "Learning", a child is considered to be developmentally on track if they can accurately follow simple instructions and execute tasks independently when provided with guidance [23,35].

The ECDI is subsequently computed as the proportion of children who demonstrate developmental progress in at least three out of the four aforementioned domains. Therefore, the ECD status of child *i* that satisfies 3 out of 4 domains *r* can be defined as

$$ECD_i = \begin{cases} 1; if \sum_{r=1}^{4} d_{ir} \geq 3 \\ 0; Otherwise \end{cases}$$

Here $d_{ir}$ denotes whether child i is developmentally on-track in case of domain r. A child is considered to be on track in early childhood development (coded as 1) if they demonstrate progress in at least three of the assessed domains. Children who do not meet this criterion are classified as not on track (coded as 0).

**Independent variables.** This research incorporates a range of variables to elucidate the determinants and assess disparities in ECD. To identify these determinants, the study draws on previous literature [8,24,27,36–39] when selecting independent variables as showed in Supplementary Table 1 (S1 Text).

A total of 11 variables are selected, encompassing spatial, sociocultural, economic, demographic, and factors related to mother and children, which are: age of child (36–47 months, 48–59 months); sex (female, male); nutritional status of the child (severely malnourished, moderately malnourished, nourished); attendance to early childhood education (yes, no); early childhood diseases (yes, no); whether the child has books at home (yes, no); type of residence (rural, urban); the division in which the child is living (Chittagong, Dhaka, Khulna, Mymensingh, Rajshahi, Rangpur, Sylhet, Barisal); level of education of mother (pre-primary or none, primary, secondary, higher secondary); multiple childcare involvement of mother (yes, no); if the child had improper supervision (yes, no) and the wealth index quintile (poorest, second, middle, fourth, richest) of the household in which the child is living.

This study has integrated a construct, namely multiple childcare involvement or responsibility of the mother or caregiver into the determinant factors of ECD. When a mother or caretaker is involved in caring for multiple other children, then the child gets less caretaker-child involvement which in turn affects early childhood development [39]. In case of inadequate individualized care in the formative years, a child's development, health, personality, and cognitive ability are all severely affected [37]. Although maternal resources for care and the mother's mental health, nutrition, and education have been widely studied as a determinant of ECD, limited studies have empirically investigated the association of childcare involvement of mother or caretaker as an ECD factor [36,40–42]. Understanding the role of mothers' childcare responsibilities can provide valuable insights into the dynamics of caregiving and its implications for ECD outcomes. In this study, this variable is coded as yes if the mother or caretaker is looking after another child. We have also included improper supervision as it can pose significant risks to a child's physical and cognitive development, especially in disadvantaged settings [8,24,37]. Following the previous studies [8], we constructed early childhood diseases as an explanatory variable. If the child's mother or caregiver responded that the child had at least two of the certain symptoms (such as fever, diarrhoea, or signs of acute respiratory infection), then the child is considered to have early childhood diseases.

To capture the wealth inequality dimension, we have created two groups, non-poor and poor by utilising the wealth index quintile of the dataset. Using data on the ownership of products, housing features, water, sanitation, and other assets/durables that reflect the household's wealth, MICS utilized principal components analysis to construct the wealth index quintile [43]. It gives the households an economic ranking and is employed in the literature as an equivalent for household statistics on wealth, income, or consumption when attempting to quantify living standards [44,45]. Following previous studies [46–49], we have re-grouped the bottom two quintiles (poorest and poorer) as the poor group and the upper three quintiles (middle, richer and richest) as the non-poor group. Our final sample consisted of 9,442 children, of which 4406 (46.66%) belonged to the poor group and 5036 (53.34%) belonged to the non-poor group.

**Empirical strategy**

As the purpose of this study is to explore the inequality in early childhood development (ECD) status, several econometric methods have been employed for the analysis:

**Logistic regression to determine odds ratios.** To explore the factors that influence ECD status in Bangladesh, logistic regression analysis is employed to calculate the odds ratio (OR) relative to a reference category variable. This method resembles multiple linear regression, except that the response variable is binary. By leveraging the odds ratio, one can discern whether a specific exposure influences a given outcome and assess the relative significance of various factors for the same event [50]. Thus logistic regression method has been used to find the determinants of ECD in Bangladesh.

**Concentration curves and concentration indices to measure inequality.** In healthcare economics, concentration curves and concentration indices serve as an important instrument for assessing and quantifying socioeconomic inequalities in health outcomes, healthcare utilization, and access to healthcare services [51]. These measures offer valuable insights into the distributional aspects of health and healthcare across different population groups, which in turn help policymakers and researchers identify disparities and formulate targeted interventions to promote equity in health.

Concentration curves visually depict the distribution of health outcomes or healthcare utilization, across socioeconomic groups. They plot the cumulative share of the variable of interest against the cumulative share of the population, stratified by socioeconomic status. The resulting curve provides a graphical representation of the degree of socioeconomic inequality in the variable under consideration. A concentration curve that lies above or below the line of equality indicates the existence of inequality for that indicator.

On the other hand, concentration indices provide a numerical measure of socioeconomic inequality in health-related variables. They quantify the degree of concentration or disparity in the distribution of the variable across socioeconomic groups, ranging from perfect equality (0) to maximal inequality (1 or -1, depending on the direction of inequality). The concentration index is calculated as twice the area between the concentration curve and the line of equality. The magnitude of the concentration index reflects the extent of socioeconomic inequality, with larger absolute values indicating greater disparities.

If $L_h(p)$ is the Concentration curve of the health variable h, at percentile $p$ of the socioeconomic distribution (e.g., ranked by wealth), then formally, the concentration index can be defined as [51]

$$C = 1 - 2\int_0^1 L_h(p)dp$$

(1)

To facilitate computation, the concentration index can be defined more conveniently as the covariance between the health-related variable and the fractional rank in the distribution of living standards [51]:

$$C = \frac{2}{\mu}cov\,(h, r)$$

(2)

Here $\mu$ is the mean of the health variable h, and $r$ is the fractional rank of individuals by socioeconomic status, such as, wealth score. By providing both visual representations and numerical measures of inequality, these methods enhance our understanding of the distributional aspects of health outcomes.

**Standard decomposition of the concentration indices.** The concentration index on its own indicates whether inequality is present in the health variable, but it does not show the sources of inequality. Therefore, a popular approach for determining the causes of inequality is the decomposition method developed by Wagstaff et al. (2003) [52]. It is possible to break down changes in income-related inequality in health outcomes into two categories using this method: changes in the main drivers of inequality in health outcomes and changes in the health elasticities with regard to these factors.

A decomposed form of concentration index can be expressed as follows [53]:

$$C = \sum_k \frac{\beta_k \overline{x}_k}{\overline{H}} \cdot C_k + \frac{GC_\epsilon}{\overline{H}}$$

(3)

Here, the average marginal effect is represented by $\beta_k$, $\overline{H}$ is the mean of the health variable and $C_k$ is the concentration index of covariate $x_k$. The inequality explained by the covariates can be represented as $\frac{\beta_k \overline{x}_k}{\overline{H}} \cdot C_k$ and it shows how much of the inequality in health (C) is due to inequality in the explanatory variables. The residual part is represented by $\frac{GC_\epsilon}{\overline{H}}$, where $GC_\epsilon$ is the generalized Concentration Index of the residual.

## Results

### Summary statistics

The summary statistics of poor and non-poor children for different socio-economic characteristics are presented in Table 1.

For the represented sample of children aged 36–47 months, the number of poor and non-poor were 46.6% and 53.4% respectively, and for those aged 48–59 months 46.8% and 53.3%, respectively. Of the poor group, 52.16% of children were male and 47.84% were female. In the non-poor group, 51.1% were male and 48.9% were female. The majority (64.68%) of the severely malnourished children belonged to the poor group. The majority (57%) of the nourished children belonged to the non-poor group. Of the children who attended early childhood education, more than half (60.83%) belonged to the non-poor group. More than half (53%) of the children who live in rural areas are poor and most (81.07%) urban children are non-poor.

A higher proportion (61.65%) of non-poor children have books at home, which is lower (38.35%) for poor children. In Dhaka, Chittagong, and Khulna, the majority (65.8%, 60.98% and 60%) of the children belong to the non-poor group. In Barisal, Mymensingh and Rangpur, the majority (66.1%, 60.73% and 65.41% respectively) of the children belong to the poor group. Mothers whose highest level of education is secondary or higher secondary education comprised the majority (61.13% and 87.86%, respectively) of the non-poor group.

Conversely, the majority of mothers (74.98% and 65.81%) whose highest level of education is pre-primary or primary, fall into the poor category. Among the children who have improper supervision, the majority (62.56%) belong to the poor group, which is lower (37.44%) for the non-poor group. Childcare involvement of mothers is equal between the two groups.

The percentage distribution of children at different domains of Early Childhood Development (ECD) for poor and non-poor children is provided in Table 2. Of the children whose ECD is on track, the majority (53.34%) belongs to the non-poor group, compared to a lower portion (44.21%) of poor children. The highest disparity is seen in the literacy-numeracy domain, where only 34.17% of the children belong to the poor group, compared to 65.83% of the non-poor children. In the other three domains as well, non-poor children secure a higher percentage. For the physical and social-emotional

**Table 1. Summary statistics of poor and non-poor children for different socio-economic characteristics.**

| Variables | Categories | Poor | | Non-poor | | p-value |
|---|---|---|---|---|---|---|
| | | Number of children | Percentage | Number of children | Percentage | |
| Age (in months) | | | | | | 0.895 |
| | 36-47 | 2232 | 46.6 | 2558 | 53.4 | |
| | 48-59 | 2174 | 46.73 | 2478 | 53.27 | |
| Sex | | | | | | 0.321 |
| | Male | 2298 | 52.16 | 2575 | 51.13 | |
| | Female | 2108 | 47.84 | 2461 | 48.87 | |
| Nutritional Status | | | | | | <0.001 |
| | Severely malnourished | 271 | 64.68 | 148 | 35.32 | |
| | Moderately malnourished | 1243 | 53.97 | 1060 | 46.03 | |
| | Nourished | 2892 | 43.04 | 3828 | 56.96 | |
| Attendance to early childhood education | | | | | | <0.001 |
| | Yes | 698 | 39.17 | 1084 | 60.83 | |
| | No | 3708 | 48.42 | 3950 | 51.58 | |
| Early childhood diseases | | | | | | 0.128 |
| | Yes | 975 | 48.58 | 1032 | 51.42 | |
| | No | 3431 | 46.15 | 4004 | 53.85 | |
| Child has book at home | | | | | | <0.001 |
| | Yes | 1880 | 38.35 | 3022 | 61.65 | |
| | No | 2526 | 55.64 | 2014 | 44.36 | |
| Residence | | | | | | <0.001 |
| | Rural | 4074 | 52.99 | 3614 | 47.01 | |
| | Urban | 332 | 18.93 | 1422 | 81.07 | |
| Division | | | | | | <0.001 |
| | Barisal | 546 | 66.1 | 280 | 33.9 | |
| | Chittagong | 773 | 39.02 | 1208 | 60.98 | |
| | Dhaka | 618 | 34.2 | 1189 | 65.8 | |
| | Khulna | 527 | 40.11 | 787 | 59.89 | |
| | Mymensingh | 348 | 60.73 | 225 | 39.27 | |
| | Rajshahi | 509 | 49.37 | 522 | 50.63 | |
| | Rangpur | 730 | 65.41 | 386 | 34.59 | |
| | Sylhet | 355 | 44.71 | 439 | 55.29 | |
| Mother's education | | | | | | <0.001 |
| | Pre-primary or None | 941 | 74.98 | 314 | 25.02 | |
| | Primary | 1540 | 65.81 | 800 | 34.19 | |
| | Secondary | 1767 | 38.87 | 2779 | 61.13 | |
| | Higher secondary+ | 158 | 12.14 | 1143 | 87.86 | |
| Multiple childcare involvement of mother | | | | | | <0.001 |
| | Yes | 3337 | 49.64 | 3386 | 50.36 | |
| | No | 1069 | 39.32 | 1650 | 60.68 | |
| Improper supervision | | | | | | <0.001 |
| | Yes | 553 | 62.56 | 331 | 37.44 | |
| | No | 3853 | 45.02 | 4705 | 54.98 | |
| **Total** | | **4406** | — | **5036** | — | |

The descriptive statistics showed that nutritional status, early childhood education, book possession, residence, division, mother education, multiple childcare involvement of mother and improper supervision have significant differences between poor and non-poor groups.

PLOS Global Public Health

**Table 2. Percentage distribution of children at four domains of early childhood development (ECD) for poor and non-poor children.**

| Name of the domains/ indicator | Number of Poor (%) | Number of Non-poor (%) |
|---|---|---|
| Literacy- numeracy | 922 (34.17) | 1776 (65.83) |
| Physical | 4317 (46.54) | 4958 (53.46) |
| Social-Emotional | 3140 (46.05) | 3679 (53.95) |
| Learning | 3905 (45.78) | 4625 (54.22) |
| ECD | 3046 (44.21) | 3844 (55.79) |
| **Total number of children** | **4406 (46.66)** | **5036 (53.34)** |

domains, 53.46% and 53.95% of children belong to the non-poor group. In the learning domain, 54.22% of the children are non-poor.

Early Childhood Development Index (ECDI), as denoted by the percentage score of ECD and its different domains is shown in Fig 2. In Bangladesh, the value of ECDI is 72.97 among which the non-poor accounts for 40.71% and the poor group accounts for only 32.26%.

The percentage score for the domain of "literacy-numeracy" is the lowest (28.57%) among all domains of ECD. With regards to this domain, the non-poor make up 18.81%, whereas the poor make up just 9.76%. The percentage scores for the rest of the three domains: physical, social-emotional and learning are 98.23, 72.22 and 90.34 percent respectively. In all three of these domains, the number of children from the poor group is less developmentally on track compared to the children from the non-poor group.

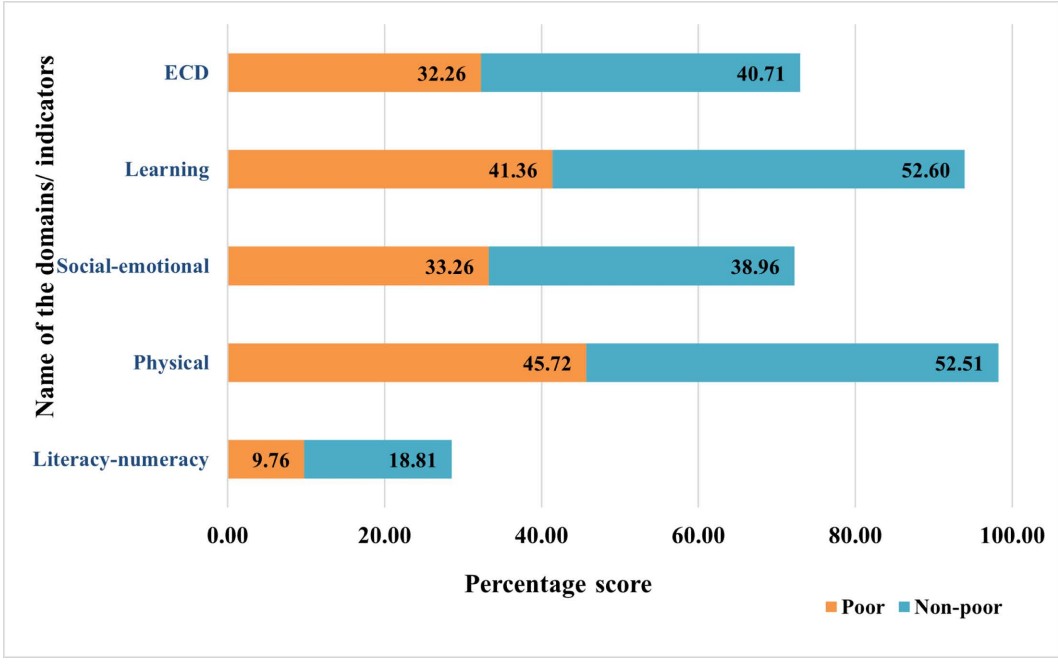

**Fig 2. Early childhood development index (ECDI) and percentage score of its different domains.**

## Logistic regression and odds ratios

Two different logistic regressions were applied to understand the determinants of ECD in Bangladesh: one for the poor group and another one for the non-poor group. The results from multicollinearity tests are provided in supplementary Table 2 in S1 Text.

The odds ratios of the determinants are represented in Table 3. Regarding the children's demographic factors, children aged 48–59 months were more developmentally on track than children aged 36–47 months. Among poor children, 65.3% attain ECD on track status upon entering the age range of 48–59 months, compared to 36–47 months. Among the non-poor children, 48–59 months olds have 90.9% higher odds of having ECD than 36–47 months. In terms of child sex, female children had higher developmentally on-track status than male children (40% in the poor group and 42.4% in the non-poor).

Compared to severely malnourished children, the odds of being developmentally on track increase by 14.3% for moderately malnourished children of the poor group, whereas it increases by 40.9% for similar children in the non-poor group. Similarly, the odds of being developmentally on track increase by 17.4% for nourished children of the poor group, whereas it increases by 48.1% for the non-poor group.

When children of poor group attend early childhood education, their status of developmentally on track increases by 48.9%. In a similar situation, the children of the non-poor group have 67.8% higher odds of being developmentally on track. If the poor children have books at home, they have 20.7% higher odds of being on track in ECD. The non-poor children, in a similar case, have 86% higher odds of being developmentally on track.

For children in poor groups, the odds of early childhood development (ECD) being on track declines by 9.4% if their mother has multiple childcare involvement compared to their counterparts who take care of a single child. Children who belong to the poor group also have notably lower (31.5%) odds of ECD being on track if they are under improper supervision than those under proper supervision.

## Concentration curves and concentration indices of health outcomes

The concentration curves for different domains of ECD are represented in Fig 3. Panel a of the figure shows the concentration curve for the domain of literacy-numeracy, while panel b, panel c and panel d show the concentration curves for physical, social-emotional and learning domains respectively.

In the above figures, it can be observed that out of four, three concentration curves are lying below the line of equality. It indicates that poor outcomes in different domains of ECD are more concentrated towards poor or disadvantaged people. In the case of the "physical" domain of ECD (Panel b), there is almost no gap between the concentration curve and the line of equality. The gap for domains of "social-emotional" (Panel c) and "learning" (Panel d) is also not as stark as in "literacy-numeracy" (Panel a). The concentration index will indicate the elements that have contributed to inequality when we break it down in the following sections. The patterns of inequality as revealed in the concentration curves also conform to the results obtained in Table 4. It shows the concentration indices for ECD as well as all four domains.

Aggregate ECD has a concentration index of 0.0413, indicating that ECD is more concentrated towards the non-poor compared to the poor group. The concentration indices also reveal that the literacy-numeracy domain has the highest inequality (0.1825) among the four domains. The concentration indices for the physical, social-emotional and learning domains of ECD are 0.0026, 0.0148 and 0.0137 respectively.

The value of concentration indices is very significant in the case of the "literacy-numeracy" domain, as well as in the case of ECD as revealed by p-values in Table 4. On the other hand, it shows moderate significance in the case of the "physical" and "learning" domains, while it is insignificant in the case of the "social-emotional" domain. The concentration curve of early childhood development is given in Fig 4. As the concentration curve is lying below the line of equality, it supports the result obtained in Table 4. The above results suggest that the poor distribution of ECD scores is more concentrated towards the low-income group or poor people.

**Table 3. Determinants of early childhood development (ECD) for poor and non-poor children.**

| Variables | Categories | Poor | | Non-poor | |
|---|---|---|---|---|---|
| | | Odds ratio (95% CI) | *p-value* | Odds ratio (95% CI) | *p-value* |
| Age (in months) | | | | | |
| | 48-59 | 1.653 (1.437-1.901) | <0.001 | 1.909 (1.643-2.217) | <0.001 |
| | 36-47 (RC) | 1.000 | — | 1.000 | — |
| Sex | | | | | |
| | Female | 1.399 (1.224-1.599) | <0.001 | 1.424 (1.241-1.633) | <0.001 |
| | Male (RC) | 1.000 | — | 1.000 | — |
| Nutritional Status | | | | | |
| | Moderately malnourished | 1.143 (.857-1.525) | 0.361 | 1.409 (.959-2.07) | 0.08 |
| | Nourished | 1.174 (.893-1.544) | 0.25 | 1.481 (1.027-2.136) | 0.036 |
| | Severely malnourished (RC) | 1.000 | — | 1.000 | — |
| Attendance to early childhood education | | | | | |
| | Yes | 1.489 (1.205-1.839) | <0.001 | 1.678 (1.358-2.073) | <0.001 |
| | No (RC) | 1.000 | — | 1.000 | — |
| Early childhood diseases | | | | | |
| | Yes | 1.119 (.844-1.484) | 0.435 | 0.846 (.631-1.136) | 0.267 |
| | No (RC) | 1.000 | — | 1.000 | — |
| Child has book at home | | | | | |
| | Yes | 1.207 (1.039-1.402) | 0.014 | 1.859 (1.609-2.148) | <0.001 |
| | No (RC) | 1.000 | — | 1.000 | — |
| Residence | | | | | |
| | Urban | 0.861 (.671-1.104) | 0.237 | 0.829 (.703-.976) | 0.025 |
| | Rural (RC) | 1.000 | — | 1.000 | — |
| Division | | | | | |
| | Chittagong | 1.622 (1.266-2.077) | <0.001 | 1.416 (1.029-1.947) | 0.032 |
| | Dhaka | 2.064 (1.576-2.703) | <0.001 | 1.874 (1.354-2.595) | <0.001 |
| | Khulna | 0.94 (0.727-1.216) | 0.639 | 0.953 (.689-1.318) | 0.77 |
| | Mymensingh | 0.877 (0.657-1.17) | 0.371 | 0.753 (.503-1.129) | 0.17 |
| | Rajshahi | 1.169 (0.898-1.521) | 0.245 | 0.996 (.705-1.407) | 0.981 |
| | Rangpur | 2.547 (1.958-3.313) | <0.001 | 1.82 (1.236-2.679) | 0.002 |
| | Sylhet | 0.845 (0.632-1.13) | 0.255 | 1.042 (.724-1.5) | 0.823 |
| | Barisal (RC) | 1.000 | — | 1.000 | — |
| Mother's education | | | | | |
| | Primary | 1.031 (0.86-1.236) | 0.745 | 0.892 (.66-1.207) | 0.46 |
| | Secondary | 1.166 (0.964-1.41) | 0.113 | 1.087(.823-1.436) | 0.558 |
| | Higher secondary+ | 1.154 (0.773-1.723) | 0.484 | 1.31 (.951-1.805) | 0.098 |
| | Pre-primary or None (RC) | 1.000 | — | 1.000 | — |
| Multiple childcare involvement of mother | | | | | |
| | Yes | 0.806 (.685-.95) | 0.01 | 0.985 (.847-1.145) | 0.846 |
| | No (RC) | 1.000 | — | 1.000 | — |
| Improper supervision | | | | | |
| | Yes | 0.685 (.562-.836) | <0.001 | 0.93 (.708-1.223) | 0.605 |
| | No (RC) | 1.000 | — | 1.000 | — |

Abbreviation: RC = Reference Category. Adopted from MICS 2019.

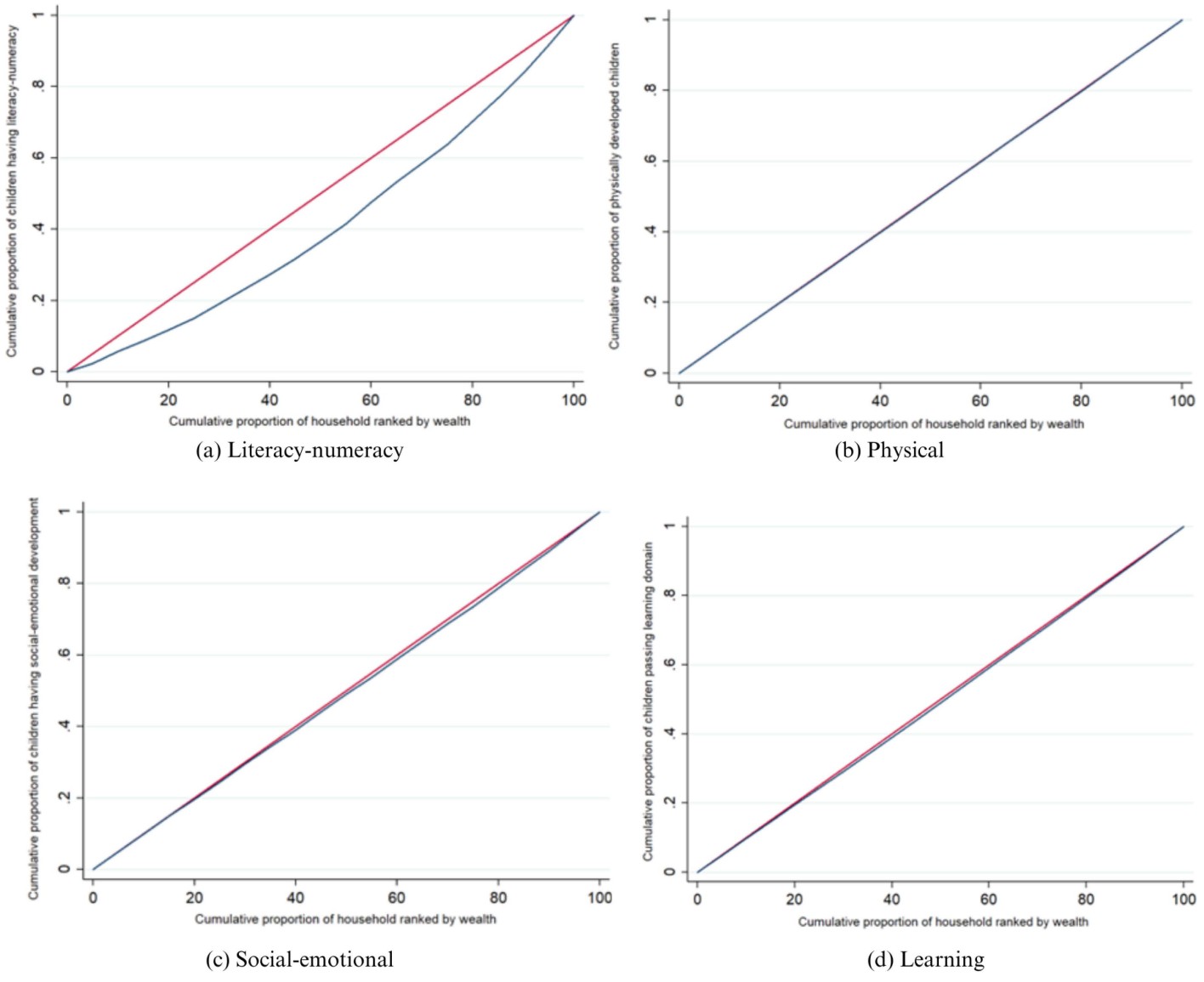

**Fig 3. Concentration curves for different domains of ECD.**

**Table 4. Concentration indices for ECD and its different domains.**

| Name of the domains/ indicator | Concentration Index | Standard Error | *p-value* | Percentage score |
|---|---|---|---|---|
| Literacy-numeracy | 0.1825 | 0.0186 | <0.001 | 28.57 |
| Physical | 0.0026 | 0.0011 | 0.024 | 98.23 |
| Social-emotional | 0.0148 | 0.0092 | 0.108 | 72.22 |
| Learning | 0.0137 | 0.0038 | 0.001 | 90.34 |
| **ECD** | **0.0413** | **0.0072** | **<0.001** | **72.97** |

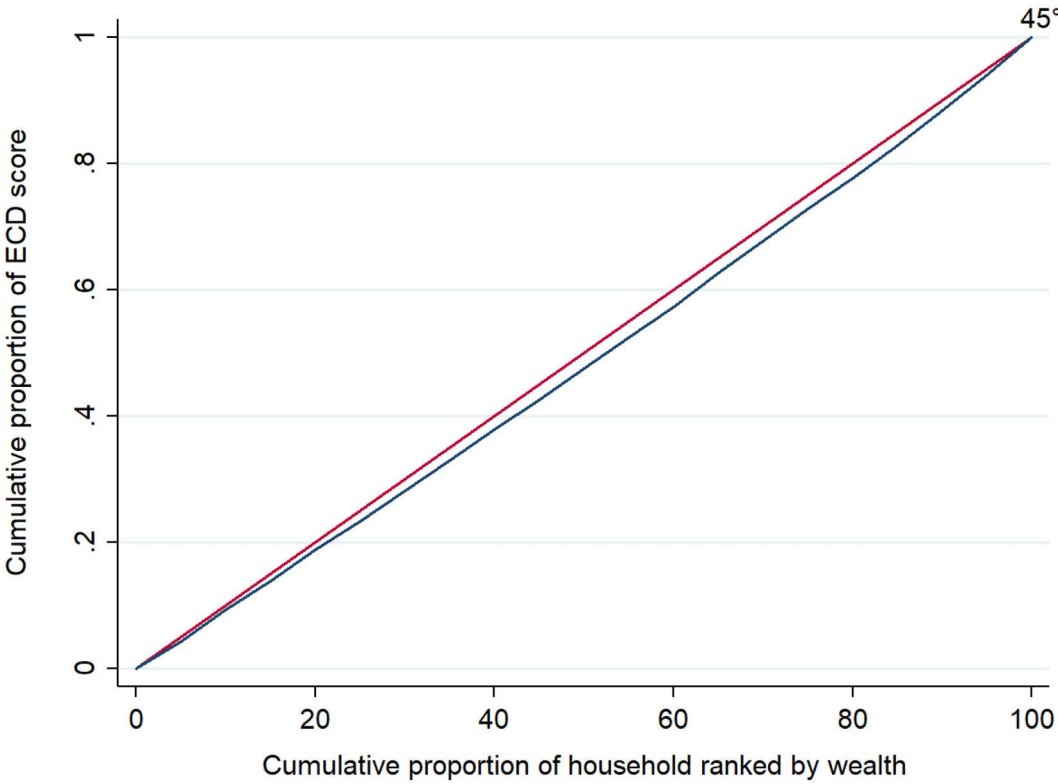

**Fig 4. Concentration curve of early childhood development (ECD).**

## Decomposition of the concentration index

The concentration index for early childhood development is decomposed in Table 5, which indicates the contributions of different factors to inequality. The table shows the findings of the elasticity analysis, the concentration index of the regressors (CI), and the absolute and percentage contribution of regressors to the inequality of ECD.

**Table 5. Decomposition of concentration index for ECD.**

| Variables | Elasticities | Concentration indices | Explained contribution | Percentage contribution |
|---|---|---|---|---|
| Age of the child | 0.6503 | -0.0003 | -0.0002 | -1.0853 |
| Sex of the child | 0.1281 | 0.0003 | 0.0000 | 0.1958 |
| Nutritional status | 0.0536 | 0.0271 | 0.0015 | 7.2352 |
| Attendance to early childhood education | 0.0181 | 0.1016 | 0.0018 | 9.1309 |
| Early childhood diseases | -0.0017 | -0.0360 | 0.0001 | 0.2968 |
| Child has book at home | 0.0525 | 0.1083 | 0.0057 | 28.2823 |
| Type of Residence | 0.0004 | 0.4324 | 0.0002 | 0.9666 |
| Division | -0.0222 | -0.0077 | 0.0002 | 0.8452 |
| Mother's education level | 0.0609 | 0.1449 | 0.0088 | 43.9067 |
| Multiple childcare involvement of mother | -0.0224 | -0.0370 | 0.0008 | 4.1231 |
| Improper supervision | -0.0055 | -0.2211 | 0.0012 | 6.1027 |

The analysis shows that mother's education level is the most dominant contributor, explaining approximately 43.9% of the observed inequality. This strong influence can be attributed to both the high elasticity (0.6069) of the mother's education in predicting ECD outcomes and its pronounced concentration among wealthier households (concentration index: 0.1449). This finding underscores the critical role of maternal education in shaping child development and highlights the existing educational disparities between rich and poor households.

The second most influential factor is whether the child has a book at home, contributing about 28.3% to the overall inequality. Books at home are a key indicator of a stimulating learning environment, which tends to be more common among affluent families. Attendance in early childhood education programs and nutritional status also make notable contributions, accounting for 9.1% and 7.2% of the inequality, respectively. These results reflect both the importance of early learning experiences and adequate nutrition in fostering early development, as well as their unequal access across wealth groups.

Additional variables, such as improper supervision and multiple caregiving involvement of mother contribute 6.1% and 4.1%, respectively, indicating that caregiving practices also play a meaningful role in explaining disparities in ECD. In contrast, variables like the sex of the child, early childhood diseases, and type of residence have minimal influence, each contributing less than 1% to the observed inequality. Interestingly, the division of residence and mother's involvement have negative concentration indices, suggesting a slight pro-poor orientation; however, their absolute contributions to inequality are relatively small.

Thus, the decomposition analysis highlights that wealth-based disparities in ECD outcomes in Bangladesh are primarily driven by differences in maternal education, learning resources at home, access to early education, and child nutrition. These findings emphasize the need for equity-focused interventions that improve access to quality education, early learning opportunities, and nutrition support for children from economically disadvantaged households.

## Discussion

The findings of this study underscore the inequality across various dimensions of early childhood development between poor and non-poor children in Bangladesh. The performance of children in the poor group is consistently lower across all four domains of early childhood development, with the most significant disparity observed in the literacy-numeracy domain. This finding is consistent with findings from related studies [24,27,28,31], which demonstrated a significant gap in ECD based on wealth inequality, including the decomposition studies [32,33]. It is also consistent with other studies that found low literacy and numeracy skills related to low household wealth status [29,54–57]. This finding can be explained by the phenomenon that children from disadvantaged families do not get access to experiences that foster the development of essential skills for literacy and numeracy, including phonological awareness, vocabulary, stimulation, oral language and counting [54,57–59].

The relatively small concentration indices were observed for the physical, socio-emotional, and learning domains of ECD in Bangladesh. One possible explanation is that these developmental domains are more strongly shaped by biological and universal processes, such as physical growth and emotional attachment, which occur regardless of socio-economic background [60,61]. Additionally, common caregiving practices, such as nurturing, play, and basic physical interaction, may be more evenly distributed across income groups [62]. As a result, these domains may be less sensitive to differences in wealth or access to resources, in contrast to the literacy-numeracy domain, which is directly influenced by access to educational inputs and parental literacy [63–65].

Our study identifies several pivotal factors that significantly influence ECD outcomes, differentiating between poor and non-poor groups. Attendance at ECE programs positively impacts ECD for both groups. Previous research has consistently highlighted the benefits of early education interventions [66,67]. Early childhood education supports child development by promoting biological, psychological, and emotional changes [4,68]. We found that mothers' multiple childcare involvement plays a negative role in poor children's ECD. It can be attributed to the fact that having to care for multiple

children affects the quality of individual children's interaction with mother. As mother's close interaction is very important for early childhood development, it thus affects ECD [69]. Children who have multiple siblings seem to perform worse across a range of educational and developmental domains [70–72]. Having multiple children puts financial pressure on a poor household, thus affecting the parent-child interaction, nutrition and access to ECE education and learning materials. Improper supervision also negatively affects the early childhood development of children of poor groups [73,74]. Where wealthier households can arrange for non-parental supervision, such as daycare centers, preschools, nurseries, and babysitters, low-wealth families cannot do so. Therefore, ensuring high-quality supervision for poor children is essential for optimal outcomes [75]. Maternal education level and adequate nutrition also impact ECD significantly. We found that the same intervention, such as ECE education or access to books has less positive effect on poor children, thus suggesting that poverty itself causes inequality due to multiple stressors [14,76]. Overall, we found that disparities in ECD outcomes are closely tied to household wealth status and access to educational resources, including books. These findings corroborate studies conducted in other global settings [18,77–80].

Our study has multiple policy implications. The findings underscore the urgent need for comprehensive intervention strategies. Addressing wealth disparities, promoting ECE participation, and enhancing maternal support systems are critical steps toward narrowing the ECD gap. Especially, the domain of "literacy-numeracy" needs special attention to improve the ECD score in Bangladesh. Addressing the disparities in early childhood development in Bangladesh necessitates a comprehensive and integrated approach that encompasses various key factors. Ensuring adequate coverage of interventions is crucial [6]. This involves not only implementing interventions but also ensuring that they reach all children, especially those from disadvantaged backgrounds. Governments, donors, and civil society must work together to establish and expand the reach of ECD interventions across the country. Financial commitments need to be ensured to sustain ECD initiatives in the long term [5,81]. As policy formulation plays a significant role in shaping the ECD landscape, the government needs to develop and implement policies that prioritize early childhood development and address the specific needs of disadvantaged children.

Establishing ECD centers specifically designed for lower-income children, equipped with literacy and numeracy training facilities, can help ensure that they have access to quality early childhood education and care. Investing in resources such as books and educational materials is another critical intervention strategy. Additionally, interventions should focus on improving access to Early Childhood Education (ECE) programs and promoting healthy nutrition and caregiving practices, which are essential for children's overall development. For mothers who have multiple childcare responsibilities, they need to be supported through ECD centers and daycare, so that their children do not get affected. Community involvement and partnerships with the private sector can also be beneficial for expanding access to ECD services and resources.

This study is based on secondary data from the MICS 2019 survey, which inherently limits the scope of analysis to variables available in the dataset. Important determinants of early childhood development—such as parental mental health or community-level interventions—thus may not have been captured. Additionally, the cross-sectional nature of the data restricts causal interpretations. More research is needed on the impact of interventions on children's development, particularly among disadvantaged populations. Future research should take into account longitudinal cohort studies to better understand the disparity and its drivers.

## Conclusion

This study paints a clear picture of inequality in early childhood development in Bangladesh, with a particular focus on how wealth-based disparities affect the different dimensions of ECD. Using concentration curves and indices, we provide robust evidence of the extent and nature of socioeconomic inequality in ECD. A key contribution of this study is the disaggregation of ECDI inequality into its four core dimensions, literacy-numeracy, physical, socio-emotional, and learning, which reveals that wealth-based disparities are most severe in literacy and numeracy. This level of detail, which is often overlooked in existing literature, is essential for identifying where policy action is most urgently needed.

The findings underscore the need for targeted, evidence-based interventions for low-wealth households. These should not only address poverty reduction and wealth inequality broadly, but also ensure access to early childhood education, quality learning materials, improved maternal education and child nutrition, and the promotion of nurturing home environments. Special attention should be given to strengthening literacy and numeracy skills, as doing so can significantly improve overall ECD outcomes in low-income groups of Bangladesh.

In summary, this study highlights both the challenges and opportunities for improving the developmental outcomes of disadvantaged children in Bangladesh. Through designing and providing interventions where ECDI inequalities are most pronounced, especially in foundational cognitive skills, policymakers can create a more equitable foundation that supports the holistic development of all children, regardless of their socioeconomic background.

## Supporting information

**S1 Text. Variable description and variance inflation factors (VIFs).**
(DOCX)

**S1 Data. Bangladesh MICS6 SPSS datasets 2019.**
(ZIP)

## Acknowledgments

The authors would like to thank Worldvision Bangladesh for their insightful discussions, which have greatly enriched this work. We also thank Md. Injamul Haq Methun for the constant suggestions on the statistical analysis.

## Author contributions

**Conceptualization:** Tasfia Tasneem Ahmed, Nafis Sadik.

**Data curation:** Tasfia Tasneem Ahmed.

**Formal analysis:** Nafis Sadik.

**Investigation:** Tasfia Tasneem Ahmed, Nafis Sadik.

**Methodology:** Tasfia Tasneem Ahmed, Nafis Sadik.

**Project administration:** Tasfia Tasneem Ahmed.

**Resources:** Tasfia Tasneem Ahmed.

**Software:** Nafis Sadik.

**Supervision:** Tasfia Tasneem Ahmed.

**Validation:** Tasfia Tasneem Ahmed, Nafis Sadik.

**Visualization:** Tasfia Tasneem Ahmed, Nafis Sadik.

**Writing – original draft:** Tasfia Tasneem Ahmed, Nafis Sadik.

**Writing – review & editing:** Tasfia Tasneem Ahmed, Nafis Sadik.

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
