## [Decision Letter · Decision Letter 0]

PGPH-D-25-00061

Wealth-based Inequalities in Early Childhood Development (ECD) Outcomes in Bangladesh: A Decomposition Analysis Using MICS 2019

Dear Dr. Sadik,

Thank you for submitting your manuscript to PLOS Global Public Health. After careful consideration, we feel that it has merit but does not fully meet PLOS Global Public Health’s publication criteria as it currently stands. Therefore, we invite you to submit a revised version of the manuscript that addresses the points raised during the review process.

The association between household wealth and ECD is already studied using the MICS data from Bangladesh. It is critical for the current study to stand out with a novel finding that is important to improve ECD. Please see editor's comments at the end of the letter and carefully address those along with reviewers' suggestions.

We look forward to receiving your revised manuscript.

Kind regards,

Biplab Datta, Ph.D.

Academic Editor

Additional Editor Comments (if provided):

The current study intends to assess the ECD inequality across wealth groups in Bangladesh. However, it is already known that wealth play a critical role in ECD in Bangladesh. For example, Hasan et al. (2023) using the MICS data reported that "children from affluent families had 1.32- and 1.26 times higher odds- on track than those from the poorest families." Similar findings were reported by Rahman et al. (2023), "compared to the poor counterpart, children from households with rich wealth status had better chance to attain overall ECD status, particularly in the literacy-numeracy and learning domains." As such, it remained unclear that what the novel contribution of the current paper is to the literature.

The current study estimated a concentration index and employed a decomposition method. It was stated that "a popular approach for determining the causes of inequality is the decomposition method developed by [52]". However, the cited paper of Wagstaff and Doorslaer (2003, Health Econ) has nothing to do with decomposition. Wagstaff et al. had other papers which were about decomposition. For example Wagstaff et al. (2003, JHE, https://doi.org/10.1016/S0304-4076(02)00161-6). Please carefully review the correct citation and perform the decomposition analysis correctly. Why the wealth index quintiles are part of decomposition, when the concentration index was based on wealth scores?

In general, make a careful case about why this study is contributing to the literature. Remove repetition of extant study findings. Explicitly communicate the meaning of the decomposition analysis and link those to potential policy implications.

References

- Hasan, M. N., Babu, M. R., Chowdhury, M. A. B., Rahman, M. M., Hasan, N., Kabir, R., & Uddin, M. J. (2023). Early childhood developmental status and its associated factors in Bangladesh: a comparison of two consecutive nationally representative surveys. BMC Public Health, 23(1), 687.

- Rahman, F., Tuli, S. N., Mondal, P., Sultana, S., Hossain, A., Kundu, S., ... & Hossain, A. (2023). Home environment factors associated with early childhood development in rural areas of Bangladesh: evidence from a national survey. Frontiers in Public Health, 11, 1209068.

Reviewers' comments:

Reviewer's Responses to Questions

**Comments to the Author**

1. Does this manuscript meet PLOS Global Public Health’s publication criteria ? Is the manuscript technically sound, and do the data support the conclusions? The manuscript must describe methodologically and ethically rigorous research with conclusions that are appropriately drawn based on the data presented.

Reviewer #1: Yes

Reviewer #2: Yes

2. Has the statistical analysis been performed appropriately and rigorously?

Reviewer #1: Yes

Reviewer #2: Yes

3. Have the authors made all data underlying the findings in their manuscript fully available (please refer to the Data Availability Statement at the start of the manuscript PDF file)?

Reviewer #1: Yes

Reviewer #2: Yes

4. Is the manuscript presented in an intelligible fashion and written in standard English?

Reviewer #1: Yes

Reviewer #2: Yes

5. Review Comments to the Author

Reviewer #1: This study focuses on disparities in early childhood development among different socioeconomic groups in Bangladesh. It uses concentration curves, decomposition analysis, and logistic regression analysis to identify various factors associated with ECD. The methodology and results are clearly explained in the manuscript. However, I think it is not necessary to explain what the odds ratio is.

In the discussion section, the "limitations" of this study could be mentioned as it is a secondary analysis. Also, possible mechanisms related to wealth status and ECD could be better explained, as they are the most important findings of this study.

Overall, the manuscript is well structured and presents strong, well-researched, and methodologically sound research. However, the "Introduction" and "Conclusion" sections could be made more concise.

Reviewer #2: \item Page 8, line 208: $d_{ir}$ is not defined.

\item Page 9, line 263: the variable is `binary' other than `binomial'.

\item Page 10 and 11: The notation in equations (1), (2), and (3) is not defined. For instance, it is unclear what $p$, $r$ and $h$ represent.

\item Some $p$-values are reported with excessive decimal places. They should be rounded appropriately.

\item The significance of some findings, particularly for social-emotional and physical domains, should be discussed more critically since the concentration indices are relatively small.

\item Wealth-based disparities in ECD have been identified in previous studies (e.g., reference [24] in the paper). What new insights does this study contribute beyond existing findings?

6. PLOS authors have the option to publish the peer review history of their article (what does this mean? ). If published, this will include your full peer review and any attached files.

**Do you want your identity to be public for this peer review?** For information about this choice, including consent withdrawal, please see our Privacy Policy .

Reviewer #1: **Yes: ** Md. Imteaz Mahmud

Reviewer #2: No

---

## [Editor Report · Decision Letter 1]

PGPH-D-25-00061R1

Wealth-based Inequalities in Early Childhood Development (ECD) Outcomes in Bangladesh: A Decomposition Analysis Using MICS 2019

Dear Dr. Sadik,

Thank you for submitting your manuscript to PLOS Global Public Health. After careful consideration, we feel that it has merit but does not fully meet PLOS Global Public Health’s publication criteria as it currently stands. Therefore, we invite you to submit a revised version of the manuscript that addresses the points raised during the review process.

Having wealth index quintiles as determinant of outcomes is a serious flaw in the decomposition analysis proposed by Wagstaff et al. (2003, Journal of Econometrics). Inclusion of wealth index quintiles simply violates the key assumption, which is the basis of the decomposition. See details in the editor comments and revised the manuscript.

We look forward to receiving your revised manuscript.

Kind regards,

Biplab Datta, Ph.D.

Academic Editor

Journal Requirements:

Additional Editor Comments (if provided):

Thank you for submitting a revised version of the manuscript. It appears that the key contribution of this manuscript is to demonstrate wealth-based inequalities and decompose the role of contributing factors to the "wealth-based" inequality. However, the fundamental error in the analysis is that "wealth index quintiles" were also considered as contributing factors to "wealth-based" inequalities. The key assumption of the seminal paper on decomposition of inequality by Wagstaff et al. (2003, Journal of Econometrics) was that "everyone in the selected sample or subsample—irrespective of their income—faces the same coefficient vector" for the determinants of the outcome variable and "interpersonal variations in the outcome are thus assumed to derive from systematic variations across income groups in the determinants."

As such, inclusion of "wealth index quintiles" as determinants of the outcomes is seriously erroneous as there is no way that everyone in the sample, irrespective of their income, will have the same coefficients for wealth index quintiles. The argument stated by the authors that "how much of the observed inequality is attributable to wealth", literally makes no sense in the context of the decomposition technique. The two references cited are subject to same mistake, and unfortunately were not carefully adjudicated by the reviewers and the editors.

Please revise the analysis, following the proper methods. Please do not get misled by erroneous publications like those mentioned in the response to the review comments.

The decomposition, however, can be done for sub-groups of wealth if the authors are interested to explore. See Clarke et al. (2003, Health Economics) for details.

References:

1. Wagstaff, A., Van Doorslaer, E., & Watanabe, N. (2003). On decomposing the causes of health sector inequalities with an application to malnutrition inequalities in Vietnam. Journal of econometrics, 112(1), 207-223.

2. Heckley, G., Gerdtham, U. G., & Kjellsson, G. (2016). A general method for decomposing the causes of socioeconomic inequality in health. Journal of health economics, 48, 89-106.

3. Clarke, P. M., Gerdtham, U. G., & Connelly, L. B. (2003). A note on the decomposition of the health concentration index. Health economics, 12(6), 511-516.

This World Bank Chapter is also helpful to understand the insights:

"Explaining Socioeconomic-Related Health Inequality: Decomposition of the Concentration Index" - available at: https://www.worldbank.org/content/dam/Worldbank/document/HDN/Health/HealthEquityCh13.pdf
---

## [Editor Report · Decision Letter 2]

Wealth-based Inequalities in Early Childhood Development (ECD) Outcomes in Bangladesh: A Decomposition Analysis Using MICS 2019

PGPH-D-25-00061R2

Dear Mr Sadik,

We are pleased to inform you that your manuscript 'Wealth-based Inequalities in Early Childhood Development (ECD) Outcomes in Bangladesh: A Decomposition Analysis Using MICS 2019' has been provisionally accepted for publication in PLOS Global Public Health.

Best regards,

Biplab Datta, Ph.D.

Academic Editor